# Comparison of anti-HCV combined with HCVcAg (Elecsys HCV Duo immunoassay) and anti-HCV rapid test followed by HCV RNA analysis using qRT-PCR to identify active infection for treatment

Sitthichai Kanokudom[1,2], Kittiyod Poovorawan[3], Pornjarim Nilyanimit[1], Nungruthai Suntronwong[1], Ratchadawan Aeemjinda[1], Sittisak Honsawek[2]*, Yong Poovorawan[1,4]*

1 Center of Excellence in Clinical Virology, Department of Pediatrics, Faculty of Medicine, Chulalongkorn University, Bangkok, Thailand, 2 Center of Excellence in Osteoarthritis and Musculoskeleton, Faculty of Medicine, King Chulalongkorn Memorial Hospital, Thai Red Cross Society, Chulalongkorn University, Bangkok, Thailand, 3 Department of Clinical Tropical Medicine, Faculty of Tropical Medicine, Mahidol University, Bangkok, Thailand, 4 Fellow of the Royal Society of Thailand (FRS [T]), The Royal Society of Thailand, Sanam Sueapa, Dusit, Bangkok, Thailand

* sittisak.h@chula.ac.th (SH); yong.p@chula.ac.th (YP)

**Data Availability Statement:** All relevant data are within the manuscript.

## Abstract

Hepatitis C virus (HCV) infection can cause acute and chronic hepatitis, leading to liver cirrhosis and hepatocellular carcinoma. The World Health Organization aims to eliminate viral hepatitis by 2030 through extensive screening and treatment. To achieve this goal, comprehensive and widespread screening is essential for diagnosis and treatment. This study aims to evaluate the diagnostic sensitivity and specificity of the Elecsys® HCV Duo immunoassay (Duo-assay), which simultaneously detects anti-HCV antibodies (Duo/anti-HCV) and HCV core antigen (Duo/HCVcAg) in a single sample, compared with initially antibody testing followed by quantitative real-time polymerase chain reaction (qRT-PCR). Additionally, this study aimed to evaluate a relationship between Duo/HCVcAg and qRT-PCR assay in different genotypes. A total of 769 plasma samples were tested using the Duo-assay to further evaluate the test's performance and conduct Duo/HCVcAg correlation analysis using qRT-PCR for each genotype. Among the active infection group (anti-HCV+/RNA+; n = 473), the Duo-assay showed 100% sensitivity for detecting Duo/anti-HCV and 70.6% for Duo/HCVcAg. In the resolved infection group (anti-HCV+/RNA–; n = 176), the assay showed 100% sensitivity for Duo/anti-HCV and 100% specificity for Duo/HCVcAg. In the non-infected group (anti-HCV–/RNA–; n = 120), the assay showed 100% specificity for both Duo/anti-HCV and Duo/HCVcAg. Moreover, no correlation was observed between the Duo/HCVcAg and HCV RNA tests, irrespective of genotype. These findings indicate that the Duo-assay is highly sensitive for detecting anti-HCV and specifically identifies patients with active infection. Nevertheless, cases with anti-HCV+/HCVcAg–results should undergo additional confirmation with western

**Funding:** This work was partially supported by the Roche Diagnostics (Thailand) Ltd., the Center of Excellence in Clinical Virology, Chulalongkorn University, the King Chulalongkorn Memorial Hospital and the Second Century Fund (C2F) of Sitthichai Kanokudom, Chulalongkorn University. The funders had no role in study design, data collection and analysis, decision to publish, or preparation of the manuscript.

**Competing interests:** The authors have declared that no competing interests exist.

blot/immunoblot and qRT-PCR to ensure diagnostic accuracy, especially in Blood donation facilities.

## Introduction

Hepatitis C virus (HCV) infection can cause acute and chronic liver diseases, potentially leading to liver cirrhosis and eventually the possible development of liver cancer [1]. The World Health Organization (WHO) recently estimated that 58 million people worldwide are infected with HCV, an 18% decrease from 71.1 million people reported in 2015 [2, 3]. Currently, there is no effective vaccine against hepatitis C. In 2016, the WHO announced a global HCV elimination strategy with the ambitious aim of screening at least 90% of the world's population, with >80% of those testing positive for HCV expected to be treated, hopefully resulting in a 65% reduction in liver-related mortality by 2030 [2, 4].

Beginning in 2018, several countries pledged to implement measures and campaigns to reach the WHO goal of HCV elimination by 2030, including Thailand [5]. This pledge included a test-to-treat strategy involving a rapid diagnostic test (RDT) as the first step, followed by verification of active infection using a qualitative real-time polymerase chain reaction (qRT-PCR) assay for HCV RNA. Our previous study was conducted in 11 districts of Phetchabun Province, Thailand, between 2019 and 2022 and initially identified 323,672 individuals using a RDT. The predominant genotype among the patients in Phetchabun Province was 6f, followed by 3a and 1a [6, 7]. A total of 11,676 (5.8%) individuals tested positive for anti-HCV. Among these subjects, 75% (4,157/5,527) were positive for HCV RNA, confirming an active HCV infection. Approximately 300 of these patients are currently undergoing treatment [6].

To eliminate HCV in accordance with the WHO's policy, screening and treatment of individuals are necessary. This approach would reduce the transmission of HCV to uninfected individuals and reduce the number of cases that progress to HCV-related liver disease. Antibody testing is the standard diagnostic procedure for HCV infection in blood donors, individuals with acute or chronic infection, and those who have a resolved infection. The standard two-step process for detecting active HCV infection includes an initial anti-HCV test followed by a confirmatory HCV RNA test, which can be both time-consuming and costly [3]. The WHO has subsequently approved the use of HCV core antigen (HCVcAg) tests alone as an alternative method for the diagnosis of active infection, reserving the use of HCV RNA testing for treatment decisions with direct-acting antivirals (DAAs). HCVcAg testing using Architect showed high concordance with HCV RNA detection in several studies, ranging from 89.7% to 95% [8]. Following this, the Thailand's National Health Security Office (NHSO), Ministry of Public Health (MOPH) of Thailand and other partners implemented WHO-based guidelines for diagnosis and treatment for HCV patients [9–11]. The approach follows a test-to-treat strategy using a two-step process [6, 9]. However, only the isolated HCVcAg test from Architect allow to use in Thailand. Therefore, it is essential to evaluate alternative tests to determine their effectiveness for HCV diagnosis.

In 2022, the newly developed commercial Elecsys® HCV Duo immunoassay (hereafter referred to as Duo-assay) (Roche Diagnostics GmbH, Mannheim, Germany) is designed to detect early infection (window phase) in individuals who are negative for antibodies against HCV and to confirm active infection. The assay consists of two test modules: one designed to detect HCVcAg using monoclonal antibodies targeting the cAg, and another module designed to detect anti-HCV using synthetic peptides and a recombinant protein representing the core,

NS3, and NS4 antigens to identify anti-HCV. The Duo-assay enables the simultaneous detection of Duo/anti-HCV and Duo/HCVcAg in a single specimen [12, 13]. Previously, the Duo-assay was evaluated in a multicenter study in Germany with regard to detecting HCV infection in blood donors (n = 20,634) and routine samples (n = 2,531). The results were compared with those of eight commercially available anti-HCV tests, followed by confirmation using nucleic acid testing (NAT), immunoblotting, or qRT-PCR. The Duo-assay demonstrated a specificity of >99.9% among blood donors and routine samples and a sensitivity of 99.6% among positive samples [13]. However, the efficacy for evaluating active/resolved infection in independent clinical samples is limited.

Additionally, our previous study demonstrated that genotypes 3a and 6 reduced the sensitivity of HCVcAg detection using the Architect HCVcAg assay compared to gold-standard qRT-PCR [14]. Another study also observed a positive correlation between HCVcAg and qRT-PCR in genotype 1b [15]. Currently, there are no reports elsewhere evaluating the performance of the Duo assay for HCVcAg of different genotypes.

The primary objective of this study was to evaluate the diagnostic sensitivity and specificity of the Duo-assay with regard to Duo/anti-HCV and Duo/HCVcAg compared standard anti-HCV testing followed by qRT-PCR confirmation among actively infected or resolved infected samples, and blood donor samples with HCV-negative. The study also examined whether detection of Duo/HCVcAg using the Duo-assay was suitable as an alternative to qRT-PCR. We also performed the correlation between Duo/HCVcAg and HCV RNA viral load and also deeply investigated whether different genotypes of HCV affect the performance of Duo/HCVcAg detection.

## Materials and methods

### Study design and sample selection

The study was carried out between February and April 2024 at the Center for Excellence in Clinical Virology, Chulalongkorn University, Bangkok. The study protocol was approved by the Institutional Review Board of the Faculty of Medicine, Chulalongkorn University (IRB Number 693/66) and conducted in accordance with the principles of the Declaration of Helsinki. All data used in the final analysis of this study were fully anonymized to protect participant privacy. As a result, no individual informed consent was required. The anonymized data can be made available upon reasonable request to the corresponding author, ensuring compliance with ethical standards.

We obtained the leftover plasma samples from a previous study [9], "Elimination of Hepatitis Virus in Thailand (Micro-elimination Model) Phetchabun Province Model Aiming for clearance Hepatitis virus by 2030" (IRB 023/63), to evaluate the performance of the Duo-assay in this study. In the previous study, these samples were collected from patients aged 35–64 years and were subjected to HCV antibody screening using the Bioline™ HCV RDT (Abbott Korea Inc., Gyeonggi-do, Korea) (Fig 1). Samples with negative anti-HCV RDT (non-reactive) were excluded, whereas positive anti-HCV RDT samples (reactive) were subsequently confirmed as positive for HCV RNA by qRT-PCR using the COBAS AmpliPrep/COBAS TaqMan HCV Test, v2.0 (Roche Molecular Systems, Pleasanton, CA). Consequently, samples were classified into two groups according to the results of anti-HCV RDT and HCV RNA including samples tested positive for anti-HCV RDT and HCV RNA were classified as the active infection group (n = 478), while those that tested positive anti-HCV RDT and negative for HCV RNA were categorized as the resolved infection group (n = 176). Additionally, leftover donated plasma from another previous study [16], which tested negative for anti-HCV antibody using an enzyme-linked immunosorbent assay (ELISA) and HCV RNA negative by qRT-PCR using

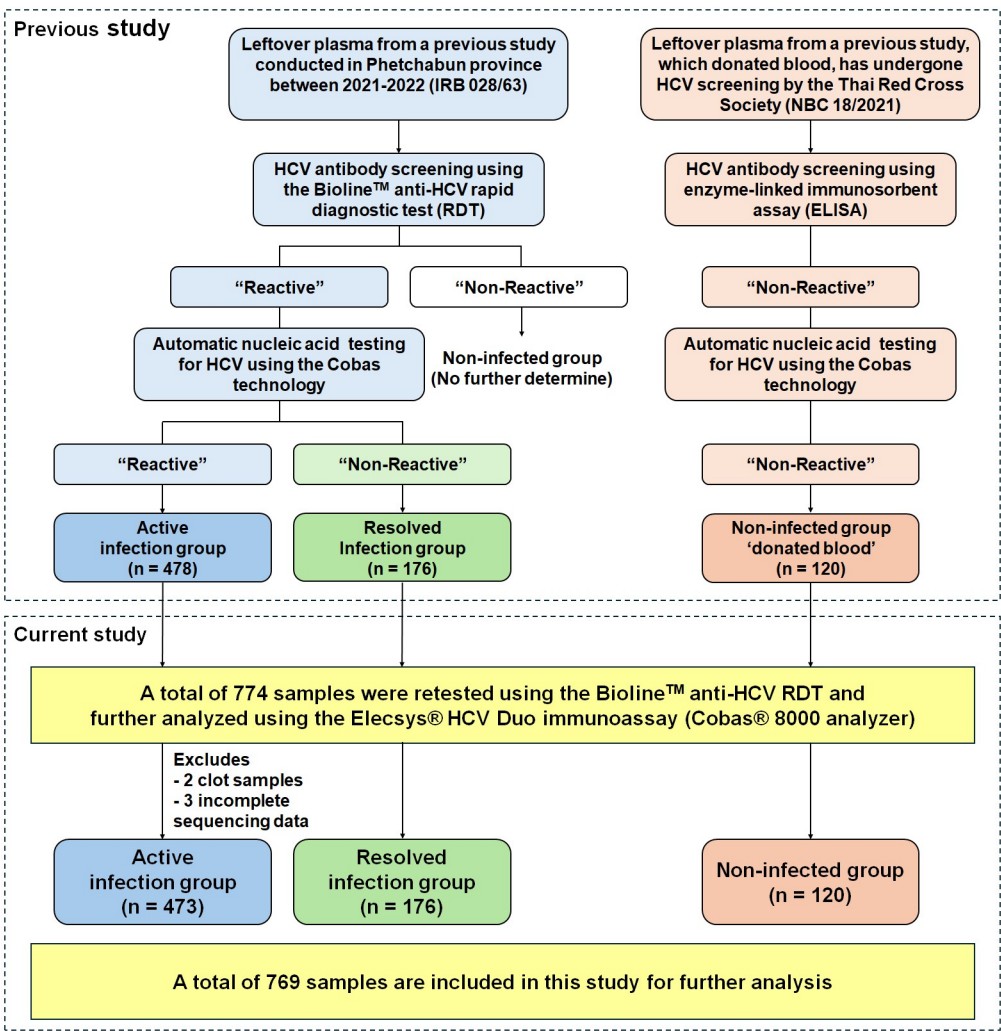

**Fig 1. Flow diagram illustrating the use of leftover plasma samples to detect HCV antibodies and antigens using the Elecsys® HCV Duo immunoassay.** Five samples from the infected group were excluded because two samples had clotted and could not be analyzed using the COBAS analyzer, and three samples had no complete sequence data. A total of 769 samples were included in this study for further analysis.

the COBAS, was obtained as a control group, referred to as the non-infected group (n = 120). All samples were done nucleic acid testing at sampling sites, and sent the left to us at 4°C. Those samples were storage at −20°C and underwent only one freeze-thaw (FT) cycle for qRT-PCR and no more than three FT cycles for antigen and antibody testing.

In the current study, a total of 774 plasma samples from previous studies were reassessed for HCV antibody using the Bioline™ HCV RDT. All samples were subjected to analyze using the Duo-assay on the Cobas® 8000 e801 analyzer (Roche Diagnostics GmbH) according to the manufacturer's instructions. Individual Duo, Duo/anti-HCV and Duo/HCVcAg results were also accessible. However, five samples were excluded due to the clotted samples and incomplete sequencing data. Thus, there were totally 769 samples included for further analysis, comprising 473 in the active infection group, 176 in the resolved infection group, and 120 in the non-infected group.

## Laboratory assessment

**Serological assay.** The Duo-assay is an electrochemiluminescence immunoassay (ECLIA) used to screen for HCV infection in serum and plasma samples. Samples were processed using the Cobas® 8000 e801 machine following the manufacturer's instructions (Roche Diagnostics GmbH). The results were reported as cutoff index (COI) values in three categories: the main result (Duo) and two modules consisting of Duo/anti-HCV and Duo/HCVcAg. The Duo result was automatically calculated from the Duo/anti-HCV and Duo/HCVcAg values, with a COI of less than 1.0 indicating a non-reactive result, and a COI of 1.0 or greater indicating reactivity. Similarly, for both Duo/anti-HCV and Duo/HCVcAg, COI values below 1.0 indicate non-reactive results, while values of 1.0 or higher indicate reactivity for anti-HCV and HCVcAg, respectively.

Furthermore, four samples from the resolved infection group showed a discrepancy between the anti-HCV RDT, which had positive results, and the Duo/anti-HCV module, which had negative results. These samples were reanalyzed using the chemiluminescent microparticle immunoassay (CMIA) of the Architect anti-HCV assay with an Architect PLUS i1000SR analyzer (Abbott GmbH, Wiesbaden, Germany) according to the manufacturer's instructions. The results were interpreted as reactive for Architect HCVcAg when equal to or greater than 1.0 signal-to-cutoff ratio (S/C), whereas results below 1.0 S/C were considered to indicate non-reactive samples.

**Quantitative HCV RNA test.** Duo/anti-HCV positive and HCV RNA positive plasma samples were subjected to qRT-PCR analysis using the Cobas® 4800 system (Roche Diagnostics) according to the manufacturer's instructions, as previously described [14]. This platform enables quantification of HCV RNA viral load for genotypes 1–6, with a lower detection limit of 15 to $10^8$ IU/mL.

**HCV genotyping.** Briefly, HCV RNA was extracted from Duo/anti-HCV positive and HCV RNA positive plasma samples using a QIAamp Viral RNA Mini kit (Qiagen, Hilden, Germany), and cDNA was subsequently synthesized using a cDNA using ImProm-II™ Reverse Transcription system (Promega, Madison, WI). PCR of the partial core gene was conducted in two groups utilizing two primer sets, following a previously published protocol [14]. The resulting PCR products were subjected to Sanger sequencing (1st Base Laboratory, Selangor, Malaysia) and subsequently genotyped based on BLASTN search results (http://www.ncbi.nlm.nih.gov).

## Data analysis

The results of diagnostic testing using the Duo-assay and qRT-PCR (as a reference standard) were compared and used to evaluate assay performance in terms of sensitivity and specificity, considering positive samples and non-infected samples, respectively. In this study, performance was evaluated using the MedCal Software Ltd. Diagnostic Test Evaluation Calculator, version 22.026 (https://www.medcalc.org/calc/diagnostic_test.php).

Linear regression analyses, correlations ($R^2$), and statistical analyses of differences between Duo/HCVcAg in terms of COI and logarithmic HCV RNA in terms of IU/mL were performed using GraphPad Prism 10 software (GraphPad Software, Boston, MA). The normality of the distribution of log HCV RNA values (in IU/mL) between reactive and non-reactive HCVcAg samples was assessed using the Kolmogorov-Smirnov test. Because the data were not normally distributed, the non-parametric Mann-Whitney $U$ test was used to compare logarithmic levels of HCV RNA between HCV Ag reactive and non-reactive samples. For normally distributed samples, the parametric Welch $t$-test was used instead. A $p$-value $<0.05$ was considered statistically significant.

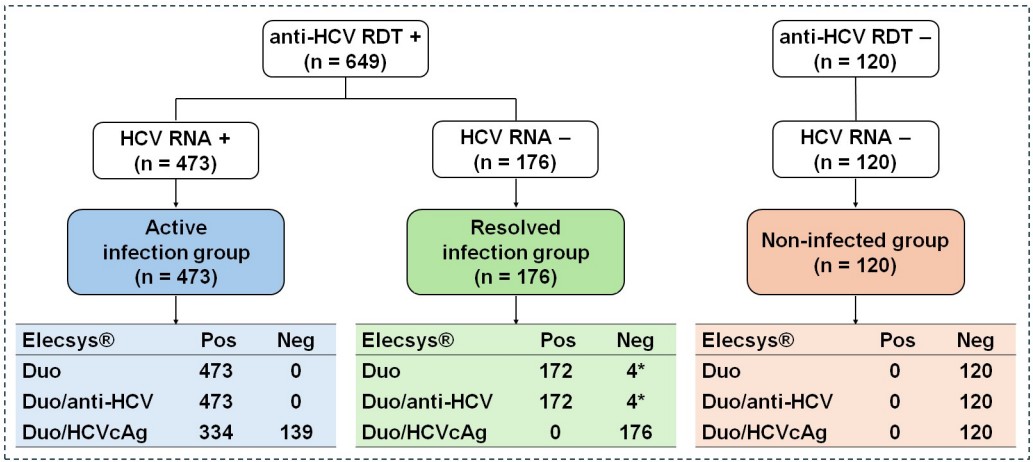

**Fig 2. Performance of the Elecsys® HCV Duo immunoassay for detecting anti-HCV and HCVcAg in plasma compared to a two-step process using anti-HCV RDT followed by qRT-PCR.** A total of 774 samples were retested in this study using the RDT. The 769 samples analyzed based on the previous HCV RNA report were further stratified into active infection, resolved infection, and non-infected groups. These samples were analyzed using the Elecsys® HCV Duo immunoassay. The results were reported as Duo, Duo/anti-HCV and Duo/HCVcAg, respectively. "Pos" indicates a positive or reactive test, whereas "Neg" indicates a negative or non-reactive test. *Samples tested positive with a faint band on anti-HCV RDT but were confirmed negative using the Architect anti-HCV assay.

## Results

### Reactivity of plasma samples for Duo/anti-HCV and Duo/HCVcAg

A total of 769 plasma samples were analyzed using the Duo-assay. The results indicated that 473 samples from the active infection group were reactive in the Duo and Duo/anti-HCV (100% sensitivity). Of these 473 samples, 334 (70.6% sensitivity) were reactive for Duo/HCVcAg, whereas 139 (29.4%) were non-reactive.

In the resolved infection group (n = 176), 172 samples were reactive in the Duo/anti-HCV, whereas 4 samples were non-reactive. Additionally, all samples in this group were non-reactive in the Duo/HCVcAg. To additionally validate the sensitivity of the Duo/anti-HCV assays in the resolved infection group, 4 non-reactive samples were retested using the Architect anti-HCV assay. The results ultimately confirmed that these 4 samples were either non-reactive for Duo/anti-HCV or false positives by RDT.

Concurrently, all 120 samples in the non-infected group were non-reactive in the Duo, Duo/anti-HCV, and Duo/HCVcAg tests (100% specificity) (Fig 2).

These findings underscore that the Duo-assay achieves 100% sensitivity for detecting anti-HCV in individuals with active infections. It has approximately 70% sensitivity for detecting HCVcAg, suggesting that nearly 30% of active infection cases were underestimated. Moreover, the Duo-assay showed 100% sensitivity in both the Duo and Duo/anti-HCV and demonstrated 100% specificity for Duo/HCVcAg in individuals with resolved infections.

### No correlation observed between Duo/HCVcAg and quantitative HCV RNA

To identify that the detection of Duo/HCVcAg was able to quantitate an amount of HCV. A total of 473 samples from active infected individuals (anti-HCV+, HCV RNA+) were subjected to HCV genotyping, and the results are described in Table 1. The four most common genotypes in the Phetchabun cohort in this study were 6f, accounting for 41.6%, followed by 3a at

**Table 1. Proportion of HCV genotyping of the study samples.**

| Total number of samples: 769 | Number (%) |
|---|---|
| Active infection group (anti-HCV+, HCV RNA+), n = 473 | |
| • Genotype 1a | 66 (14.0) |
| • Genotype 1b | 34 (7.2) |
| • Genotype 3a | 150 (31.7) |
| • Genotype 3b | 10 (2.1) |
| • Genotype 6f | 197 (41.6) |
| • Genotype 6i | 8 (1.7) |
| • Genotype 6n | 7 (1.5) |
| • Genotype 6v | 1 (0.2) |
| Resolved infection group (anti-HCV+, HCV RNA−), n = 176 | N/A |
| Non-infected group (anti-HCV−, HCV RNA−), n = 120 | N/A |

N/A, no data available due to the absence of HCV RNA.

31.7%, 1a at 14.0%, and 1b at 7.2%. By contrast, there was no available data regarding genotypes of the group negative for HCV RNA (n = 296).

No significant correlation was observed between Duo/HCVcAg results and HCV RNA viral load among genotypes 1a, 1b, 3a, and 6f ($R^2$ ranging from 0.067 to 0.212) (Fig 3A–3D). In terms of test sensitivity, the Duo/HCVcAg was able to detect genotypes 1a (80.3%), 1b (76.5%), 6f (71.6%), and 3a (62.0%) (Table 2). We also compared the HCV RNA viral load between Duo/HCVcAg reactive and non-reactive samples. Duo/HCVcAg reactive samples had significantly higher titers than non-reactive samples across genotypes 1a, 1b, 3a, and 6f ($p < 0.05$) (Table 2). These data suggest that the sensitivity of the Duo/HCVcAg assay is affected by the level of HCV RNA present in plasma samples. Our results suggest that the level of Duo/HCVcAg detection was not significantly correlated with the amount of HCV RNA. This indicates that the Duo/HCVcAg module is not suitable for the quantitative determination of HCV RNA.

## Discussion

The main purpose of this study was to evaluate the performance of one-step antibody and antigen tests for the screening and diagnosis of HCV infection in unidentified samples. The Duo-assay, which includes an anti-HCV module, exhibited 100% sensitivity for samples from the active infection group (RDT+/PCR+). Of these, 70.6% were reactive in the Duo/HCVcAg module, in contrast to previous studies reporting a sensitivity of 95.7% for the automated Architect HCVcAg assay, designed specifically for HCVcAg detection in chronic hepatitis C patients, including those co-infected with HIV or hepatitis B virus (HBV) [17].

In the resolved infection group (RDT+/PCR−), 172 of 176 samples showed concordance with the sensitivity of the two-step reference standard, which combines RDT followed by qRT-PCR analysis. However, the results for four samples were inconsistent. These samples were then re-analyzed using a different automated diagnostic test, the Architect Anti-HCV assay. The results indicated that the RDT had produced false-positive results for anti-HCV. Among 172 resolved infection samples, there was 100% agreement with the Duo-assay: 100% were positive for anti-HCV, and 100% were negative for Duo/HCVcAg, corresponding to the two-step RDT+/PCR− results. The automated analysis results were more accurate than those interpreted visually using RDTs [18]. This incidence was supported by previous studies

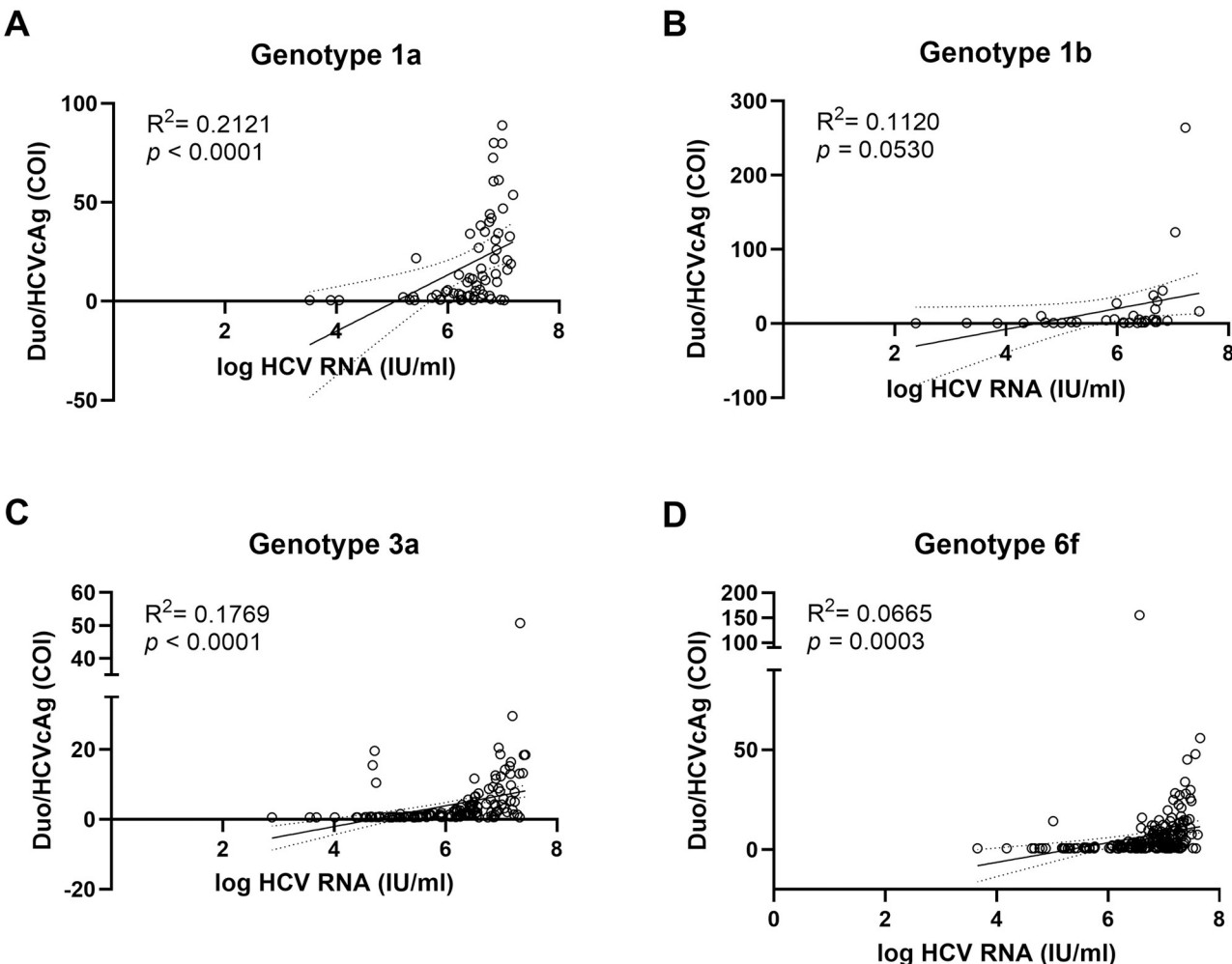

**Fig 3. Correlation between log HCV viral load and the Duo/HCVcAg.** Scatter plot illustrates the relationship between Duo/HCVcAg module, monitored based on the automated Duo/HCVcAg in COI, and log HCV RNA level in IU/mL across different genotypes, including 1a (A), 1b (B), 3a (C), and 6f (D). Correlations ($R^2$) were calculated from trendlines. Statistical significance was set at $p < 0.05$.

showing that seven manufactured anti-HCV RDTs (Alere Trueline, Benesphera, AccuTest, Abon, Blue Cross, Healgen, and SD Bioline) exhibited lower sensitivity, ranging from 65.3% to 83.0%, when compared with automated CMIA using the Architect anti-HCV assay [18, 19].

All resolved infection samples tested negative in the Duo/HCVcAg module. In the blood donor group (RDT−/RNA−), all samples were negative in the Duo, Duo/anti-HCV, and Duo/HCVcAg modules (100% specificity). Based on these collective results, we concluded that the Duo and Duo/anti-HCV modules exhibited 100% sensitivity throughout the cohort for anti-HCV+ samples, the active infection and resolved infection groups, and 100% specificity with HCV confirmed-negative samples. These results were consistent with those of a previous study conducted in Germany, which demonstrated a sensitivity of up to 99.9% among known HCV confirmed-positive samples and specificity of 99.6% among blood donor samples [12, 13]. Moreover, our data suggest that the German study calculated the Duo-assay results based solely on the anti-HCV module [12, 13]. However, our data indicates that approximately 30% of active infection group may be missed and incorrectly classified as resolved infection group

**Table 2. Comparison between Duo/HCVcAg and viral load for 447 samples from the active infection group, including genotypes 1a, 1b, 3a, and 6f.**

| Genotype | Module | Interpretation | Number | Log median [IQR] (min-max) in IU/mL | p-value* |
|---|---|---|---|---|---|
| 1a | Duo/HCVcAg | Reactive | 53 | 6.63 [6.35–6.9] (5.20–7.18) | 0.0141 |
| | | Non-reactive | 13 | 5.88 [4.69–6.37] (3.52–7.02) | |
| 1b | Duo/HCVcAg | Reactive | 26 | 6.50 [5.98–6.71] (4.63–7.47) | 0.0068 |
| | | Non-reactive | 8 | 4.59 [3.44–5.84] (2.38–6.36) | |
| 3a | Duo/HCVcAg | Reactive | 93 | 6.52 [6.21–6.99] (4.69–7.43) | <0.0001 |
| | | Non-reactive | 57 | 5.30 [4.81–5.92] (2.88–7.33) | |
| 6f | Duo/HCVcAg | Reactive | 141 | 7.00 [6.61–7.24] (5.02–7.66) | <0.0001 |
| | | Non-reactive | 56 | 6.18 [5.41–6.99] (3.66–7.58) | |

*$p$-value comparison of HCV RNA viral load between Duo/HCVcAg reactive and non-reactive samples.

(i.e., lower accuracy). Despite this limitation, to ensure accurate monitoring of treatment response in some patients, samples that test positive for anti-HCV but negative for HCVcAg should undergo follow-up an alternative HCV RNA testing to confirm the infection status.

We were the first to report the influence of genotypes on the performance of the Duo/HCVcAg assay and its relationship with viral load. This study revealed found no correlation between Duo/HCVcAg and HCV viral load corresponding to genotypes 1a, 1b, 3a and 6f when using ECLIA, although we monitored the correlation between cAg levels in COI and HCV viral load (with a median of >4.5 log IU/mL) among actively infected individuals. This is because the Duo-assay is designed for *in vitro* qualitative interpretation. Therefore, we suggest that the COI scale determined using the Duo-assay should not be used to calculate the correlation with HCV viral load. Unlike previous studies demonstrating a correlation between cAg level and HCV RNA viral load, this discrepancy arises from the design of the CMIA of the Architect HCVcAg test specifically for quantitative analysis, with the result reported in fmol/mL [14]. Other recent studies have suggested that detecting HCVcAg using the Architect assay provides high sensitivity for samples with a viral load >10,000 IU/mL [20] and that cAg can be detected within 12–15 days after infection [21]. Moreover, another study observed that 32 of 145 actively infected patients tested positive for a viral load ≥1,000 IU/mL (high titer), whereas 113 of 145 tested positives for a viral load <1,000 IU/mL (low titer). When comparing the positivity between viral load and Architect HCVcAg results showing a high correlation (r = 0.890; $p$<0.001), this study demonstrated a sensitivity of 93.9% in the high-titer group and 50.0% in the low-titer group, with a specificity of 99.1% [22].

These results were consistent with those of the current study, which showed that samples seropositive in the Duo/HCVcAg test had significantly higher levels of HCV RNA compared with non-reactive samples among patients with active infection. However, no correlation was observed between HCV RNA level and Duo/HCVcAg result, considering both reactive and non-reactive samples. Furthermore, our previous study revealed that Architect HCVcAg determined was significantly correlated with HCV RNA, regardless of HCV genotype and/or co-infection with HBV, with accuracy, sensitivity, and specificity values of 99%, 99%, and

100%, respectively [23]. Furthermore, another previous study demonstrated a strong correlation between Architect HCVcAg and HCV RNA results, with an $R^2$ value of 0.889 and $p$-value of <0.001 [19]. The Duo-assay combines two modules, including total anti-HCV antibodies that target the core, NS3, NS4A, and NS5A proteins [12], which might interfere with binding to cAg during the reaction, whereas the Architect assay was specifically designed for detecting the cAg alone.

We also explored the sensitivity of the test among different genotypes. The Duo-assay was designed using recombinant proteins representing HCV genotype 1. This was consistent with previous finding revealing that Architect HCVcAg quantification is highly correlated with genotype 1, whereas genotypes 3 and 6 exhibit lower correlations with HCV RNA levels [14]. This was consequently supported by previous studies showing that genotype 1 is commonly prevalent worldwide, accounting for 49.1%, followed by genotype 3 at 17.9%. By contrast, genotypes 5 and 6 collectively are the least common, constituting <5% of cases [24–26]. Previous studies reported that genotype 3 (particularly 3a) is highly prevalent in Southeast Asia, including Thailand, which was associated with the significant increase in narcotics use during the Vietnam War in the 1960s-1970s and unawareness regarding asymptomatic/resolved hepatitis cases associated with medical transfusions thereafter [27–29]. The current study identified genotype 6 as predominant in Phetchabun Province, which is located in the southern part of northern Thailand. Our data was concordant with those of a previous study conducted in 2018 involving a sample size of 4,769 subjects [7]. Understanding the variety of genotypes is crucial for development of effective screening processes and the use of DAAs in therapeutics, which could facilitate achievement of the goal of eliminating hepatitis C by 2030.

The results of this study support development of guidelines for identifying new patients and shorten the HCV screening process, potentially leading to more cost-effective surveillance. This can influence clinical decision-making and treatment timing, particularly in resource-limited regions where HCV RNA testing may be limited or unavailable. The results obtained from the two modules using the Elecsys® HCV Duo immunoassay (Duo-assay) can be categorized into four categories, as illustrated in Fig 4. First, if the results are anti-HCV−/cAg−, the patient is categorized as free of HCV infection; consequently, no further follow-up is needed. For the second category, if infection is diagnosed in the early stages (anti-HCV−/cAg+), known as acute hepatitis, treatment may not be immediately necessary. Instead, the patient can undergo another blood test after a few months to confirm natural clearance of the virus by the immune system. In early HCV infection, 15–45% of individuals experience spontaneous viral clearance, leading to the presence of anti-HCV antibodies and the absence of detectable HCV RNA within a few months [3, 29, 30]. Approximately 55–85% of infected individuals progress to chronic infection, characterized by the presence of anti-HCV antibodies and detectable HCV RNA [3, 29–32]. For the third category, patients with a result of anti-HCV +/cAg− (approximately 30% misclassified due to a false negative for cAg) require additional nucleic acid testing (RNA+) to determine if the infection is active, and if so, guide the appropriate course of treatment. Conversely, patients with an RNA− result do not require further follow-up. In the fourth category, patients with a result of anti-HCV+/cAg+ are deemed as needing treatment.

Some limitations were observed in this study. First, demographic characteristics were not provided due to the anonymity of the sample sites. The reference standards for evaluating a performance agreement of Duo/anti-HCV antibodies were unavailable. However, our study focused specifically on assessing the Duo-assay's ability to differentiate between individuals with active infection (anti-HCV+ and Duo/HCVcAg +) and those with resolved infection (anti-HCV+ and Duo/HCVcAg−). Another limitation is that we did not obtain samples showing RDT−/PCR+ results, which represent early infection/window-phase cases. This was

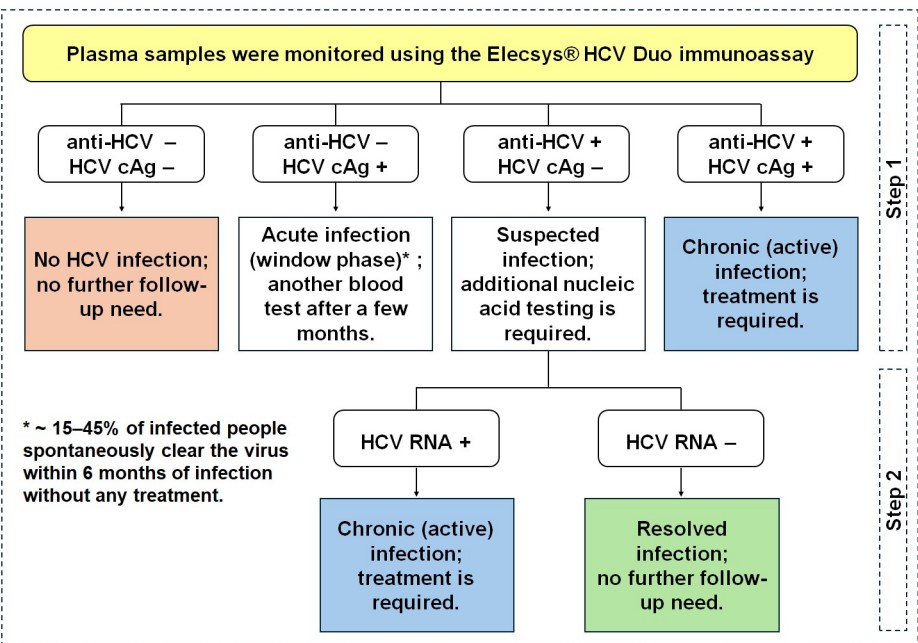

**Fig 4. Potential recommendation for the use of the Elecsys® Duo immunoassay to identify and characterize HCV status in new patients.** Duo-assay results can be interpreted according to four patterns, consisting of three sets of one-step (anti-HCV−/cAg−, anti-HCV−/cAg+, and anti-HCV+/cAg+) and one set of two-step (anti-HCV+/cAg−) processes. Samples with a result of anti-HCV+/cAg− must be confirmed using nucleic acid testing as the second step.

because the criteria used in the previous study did not include additional confirmation by PCR in the case of a negative RDT result, as this is rare [11]. Anti-HCV antibodies are typically detected after several weeks of infection.

## Conclusion

The Elecsys® HCV Duo immunoassay, which allows determination of antibodies and antigens in a single sample, is beneficial for determining HCV status. The results of this study suggest that the Duo-assay provides high sensitivity for detecting chronic HCV infection with high specificity for the diagnosis of active HCV infection. However, samples with anti-HCV+/cAg− results must be confirmed using HCV qRT-PCR or RNA detection. This tool could significantly reduce the cost of HCV RNA viral load testing, reduce turn-around time for screening and diagnostic and expedite treatment for patients. Additionally, the Elecsys® HCV Duo immunoassay could assist in achieving the goal of eliminating hepatitis C globally by 2030.

## Acknowledgments

We would like to express our gratitude to Bangpakok International Hospital for providing us with the equipment and facility for sample testing, and to all participants for their contributions and support of this project.

## Author Contributions

**Conceptualization:** Sitthichai Kanokudom, Yong Poovorawan.

**Data curation:** Sitthichai Kanokudom.

**Formal analysis:** Nungruthai Suntronwong.

**Funding acquisition:** Yong Poovorawan.

**Investigation:** Sitthichai Kanokudom.

**Methodology:** Sitthichai Kanokudom, Pornjarim Nilyanimit, Ratchadawan Aeemjinda.

**Project administration:** Sitthichai Kanokudom, Yong Poovorawan.

**Writing – original draft:** Sitthichai Kanokudom.

**Writing – review & editing:** Sitthichai Kanokudom, Kittiyod Poovorawan, Sittisak Honsawek, Yong Poovorawan.

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
