## [Decision Letter · Decision Letter 0]

24 Sep 2024

PONE-D-24-35203Comparison of anti-HCV combined with HCV cAg (Elecsys® HCV Duo immunoassay) and anti-HCV rapid test followed by HCV RNA analysis using qRT-PCR to identify active infection for treatmentPLOS ONE

Dear Dr. Poovorawan,

Thank you for submitting your manuscript to PLOS ONE. After careful consideration, we feel that it has merit but does not fully meet PLOS ONE’s publication criteria as it currently stands. Therefore, we invite you to submit a revised version of the manuscript that addresses the points raised during the review process.

Your manuscript were reviewed by two experts in the field. Although both mentioned that you manuscript is very interesting, yet many comments were generated and need to be fixed. 

We look forward to receiving your revised manuscript.

Kind regards,

Gheyath K. Nasrallah

Academic Editor

PLOS ONE

Journal Requirements:

1. When submitting your revision, we need you to address these additional requirements. Please ensure that your manuscript meets PLOS ONE's style requirements, including those for file naming. The PLOS ONE style templates can be found at https://journals.plos.org/plosone/s/file?id=wjVg/PLOSOne_formatting_sample_main_body.pdf and https://journals.plos.org/plosone/s/file?id=ba62/PLOSOne_formatting_sample_title_authors_affiliations.pdf.

 [This work was partially supported by the Roche Diagnostics (Thailand) Ltd., the Center of Excellence in Clinical Virology, Chulalongkorn University, the King Chulalongkorn Memorial Hospital and the Second Century Fund (C2F) of Sitthichai Kanokudom, Chulalongkorn University.].  

Reviewers' comments:

Reviewer's Responses to Questions

**Comments to the Author**

1. Is the manuscript technically sound, and do the data support the conclusions?

Reviewer #1: Yes

Reviewer #2: Yes

2. Has the statistical analysis been performed appropriately and rigorously? 

Reviewer #1: Yes

Reviewer #2: I Don't Know

3. Have the authors made all data underlying the findings in their manuscript fully available?

Reviewer #1: Yes

Reviewer #2: Yes

4. Is the manuscript presented in an intelligible fashion and written in standard English?

Reviewer #1: Yes

Reviewer #2: Yes

5. Review Comments to the Author

Reviewer #1: The manuscript presents a comprehensive evaluation of the Elecsys® HCV Duo immunoassay, highlighting its potential as a diagnostic tool for Hepatitis C virus (HCV) infection. The study offers valuable insights into the assay's performance, particularly in its ability to detect both HCV antibodies and core antigen. However, there are several critical aspects that need to be addressed to enhance the manuscript's clarity, the robustness of the findings, and the practical implications of the assay. Below are specific comments and concerns that should be addressed before the manuscript can be considered for publication.

Minor Comments:

1. Ensure consistent tense usage. For example, "This study aimed to evaluate" could be revised to "This study aims to evaluate" to match the present tense used throughout the abstract.

2. There are a few instances where commas are either missing or incorrectly placed. For example, in the sentence "The results were interpreted as reactive for Duo, as calculated from the highest score among anti-HCV and HCV cAg results," a comma after "Duo" seems unnecessary.

3. The manuscript might benefit from ensuring consistency in the use of italics and bold for emphasis. For instance, "HCV" is sometimes italicized and other times not. Consistency is key.

4. Ensure that all acronyms are defined at their first occurrence. For example, "ECLIA" and "COI" are used without prior definition.

5. The resolution of the figures is very poor. Consider using color coding to differentiate between the different groups in figure 1. This would help visually separate different parts of the flow and make the figure more engaging.

Major comments

1. One of the primary concerns with this manuscript is that it does not substantially add to the body of knowledge already established by the 2022 study on the Elecsys® HCV Duo immunoassay. The earlier study provided a comprehensive multicenter evaluation of the assay's specificity and sensitivity across diverse populations and settings. While your manuscript offers localized data from a Thai cohort, the overall findings align closely with those of the 2022 paper, without introducing significant new insights or advancements. To strengthen the manuscript, the authors should highlight the novelty of their study and what their study add to the exiting literature.

2. The manuscript briefly mentions the lack of correlation between HCV core antigen (cAg) levels and HCV RNA. This is an important point, but it needs a more in-depth analysis. The discussion should explore potential reasons for this lack of correlation and its implications for the use of the Elecsys® HCV Duo immunoassay as a standalone diagnostic tool. Additionally, compare these findings with the 2022 study, which did not emphasize this issue, and discuss whether this observation is novel or aligns with other emerging data in the field.

3. The manuscript explicitly states that there was no overall correlation found between HCV core antigen (cAg) levels and HCV RNA. Given this finding, the rationale for proceeding with genotyping and sequencing is unclear. The authors need to explain why they chose to perform genotyping after determining that there was no significant correlation between cAg and RNA. Was the genotyping intended to investigate whether specific HCV genotypes might influence this lack of correlation, or was there another hypothesis being tested? Without this explanation, the inclusion of genotyping and sequencing appears disconnected from the primary findings, raising questions about the necessity and relevance of these procedures within the study. This clarification is crucial for understanding the study's design and the implications of the results.

4. The manuscript should reference any existing literature that suggests genotyping might influence the correlation between cAg and RNA levels. If such references exist, they should be cited to justify the decision to include genotyping in the study. If no such references are available, the authors should clarify that their approach was exploratory and explain the reasoning behind this exploration. This clarification is crucial for understanding the study's design and the implications of the results.

5. The manuscript reports that a significant number of samples in the active infection group tested positive for anti-HCV but negative for HCV cAg. This finding is critical, as it impacts the reliability of using cAg as a standalone marker for active infection. The authors should discuss potential reasons for this discrepancy, such as assay sensitivity, low viral loads, or technical factors. They should also address the clinical implications of these findings, particularly the importance of confirming anti-HCV+/cAg- results with HCV RNA testing to avoid false negatives.

Reviewer #2: Reviewer Comments to Authors: Manuscript No.: PONE-D-24-35203

Comparison of Elecsys® HCV Duo assay versus Anti-HCV RDT followed by qRT-PCR to Identify active infection

I went through the manuscript, and after thoroughly reviewing and evaluating the scientific attributes and

level of importance of the manuscript, I am hereby sending the following comments to the Authors:

The authors’ aim was to compare the Elecsys® HCV Duo immunoassay and the traditional two-step

process, which involves initial anti-HCV antibody testing followed by quantitative real-time polymerase

chain reaction (qRT-PCR) for detecting active hepatitis C infections. Additionally, the study aimed to

evaluate the relationship between HCVcAg and qRT-PCR assay in different genotypes.

Comments to Authors:

1. While the study holds significant value, especially within the context of the global effort to

eliminate HCV, the authors should consider the following points:

A. Introduction:

i. In line 69: include a reference on the adoption of HCV core antigen testing by WHO

recommendation. This will provide credibility regarding the global significance of core antigen

testing.

ii. Provide a brief mention of the national guidelines for HCV testing in Thailand and whether

HCV core antigen testing has been adopted in these guidelines. This will help ground the study

in the context of local diagnostic practices and highlight the relevance of your findings for

Thailand’s national guidelines.

iii. Elaborate on the advantages of using the assay of anti-HCV combined with HCV cAg in real

world settings, particularly its implications for reducing costs and turnaround time in HCV

screening and diagnostics.

B. Materials and Methods:

i. Flow and Clarity: Improving the flow of the methodology would significantly enhance its

clarity and make it easier for readers to follow. A more logical structure would ensure that

each step of the study is clearly outlined, facilitating comprehension. For example, as noted

below, you could improve the transition between the previous and current study methods for

better readability.

ii. Serology assay in the Laboratory assessment Section: Reorganize this section by starting with

information on how the samples were categorized into active infection and resolved infection

groups, referencing the use of Bioline™ HCV RDT from the previous study (Figure 1). Then, follow this by explaining how the current study employed the Elecsys® Duo assay (anti-HCV & HCV cAg) to further assess these groups. This approach will make it easier for readers to understand how the two testing methods are related and integrated into the study.

iii. Sample Processing: More detail should be mentioned on how samples were processed, in

particular in regard to the handling of frozen samples and any potential effects on antigen or

RNA stability.

iv. Line 169 - Reference Standard: In line 169, it is mentioned that qRT-PCR was used as the

reference standard to evaluate the Duo-assay's performance in terms of sensitivity and

specificity. While this is correct for the HCV core antigen (HCV cAg) component of the assay,

it is not correct for the anti-HCV component. For anti-HCV testing, a more specific assay is

required for confirmation, such as the Line Immunoassay (LIA). Typically, tests like the INNO

LIA HCV Score are used to confirm the presence of HCV antibodies with greater specificity.

Therefore, it is important to clarify what reference standard was used to evaluate the

performance of the Duo anti-HCV component. If a Line Immunoassay or a similar confirmatory

test was utilized, this should be explicitly mentioned. Doing so will enhance the accuracy of

the methodology and ensure that the reported sensitivity and specificity values for the anti

HCV assay are appropriately validated against the correct reference standard.

C. Results:

i. The result part needs improvement in terms of clarity and interpretation. By Embedding brief

interpretations within the results to highlight the clinical or practical significance of the

findings, without waiting for the discussion section.

ii. Line 191 and 194: In lines 191 and 194, you refer to "the Duo and anti-HCV modules." For

clarity and consistency, I suggest changing this to "the Duo/anti-HCV modules" throughout

the manuscript wherever the Duo and anti-HCV are mentioned. This will provide a clearer and

more concise reference to the different components of the assay, making it easier for readers

to follow.

iii. Line 197: In line 197, you mention "To validate the sensitivity of the Duo and Elecsys® anti

HCV assays." If the study only used the Elecsys® Duo assay (which includes both anti-HCV and

HCV cAg components), and not a separate Elecsys® anti-HCV assay, please revise this

sentence for accuracy. Instead, it should read: "To validate the sensitivity of the Elecsys® Duo

(anti-HCV) assay." This will ensure that readers are not confused about whether multiple

assays were used, and it accurately reflects the methodology. If both the Duo and separate

anti-HCV assays were used, make sure this distinction is clearly outlined.

D. Discussion:

i. While the Elecsys® Duo assay demonstrates 100% sensitivity for detecting anti-HCV

antibodies, its 70% sensitivity for detecting HCV core antigen (cAg) indicates that 30% of

active infections might be missed if relying solely on cAg detection. This limitation should be

explicitly acknowledged in the manuscript, particularly in the context of resource-limited

settings where access to HCV RNA testing may be restricted or unavailable. In these settings,

the inability to detect all active infections could impact clinical decision-making, delaying

treatment initiation for some patients. Discussing the importance of complementary testing

(e.g., follow-up with RNA testing) where possible, or alternative strategies to manage false

negatives, will make the discussion more balanced and practical.

ii. Detailed discussions on clinical applicability will enhance the quality of the manuscript.

E. Informed consent statement:

i. The ethical considerations concerning anonymized data usage should be emphasized more.

F. Funding:

i. To further reduce any potential conflict of interest (COI) concerns, it is recommended that

you include the following statement in the Funding section: "The funders had no role in study

design, data collection and analysis, decision to publish, or preparation of the manuscript."

This statement will reassure readers and reviewers that the research was conducted

independently and that the funding sources did not influence any aspect of the study's design,

execution, or reporting. This addition will enhance the transparency of the manuscript and

ensure that any concerns regarding objectivity are adequately addressed.

G. Figures and Tables:

i. Provide a legend for each figure and table to guide the reader in interpreting the data,

especially for less familiar readers

ii. The tables should be revised to be clearer and more informative.

H. Minor Comments:

i. Review the manuscript for minor grammatical issues. Shorter, clearer sentences would

improve readability.

ii. Consistency in terminology (e.g., using "Elecsys® Duo assay" versus "Duo-assay") should

be maintained throughout the text.

6. PLOS authors have the option to publish the peer review history of their article (what does this mean?). If published, this will include your full peer review and any attached files.

Reviewer #1: No

Reviewer #2: No

---

## [Author Response · Author response to Decision Letter 0]

29 Oct 2024

Response to editor (R1)

Journal Requirements:

1. When submitting your revision, we need you to address these additional requirements. Please ensure that your manuscript meets PLOS ONE's style requirements, including those for file naming. The PLOS ONE style templates can be found at https://journals.plos.org/plosone/s/file?id=wjVg/PLOSOne_formatting_sample_main_body.pdf and https://journals.plos.org/plosone/s/file?id=ba62/PLOSOne_formatting_sample_title_authors_affiliations.pdf.

Response: We ensured that the first draft MS and revision version are provided according to the PLOS ONE's style requirements.

 [This work was partially supported by the Roche Diagnostics (Thailand) Ltd., the Center of Excellence in Clinical Virology, Chulalongkorn University, the King Chulalongkorn Memorial Hospital and the Second Century Fund (C2F) of Sitthichai Kanokudom, Chulalongkorn University.]. 

Response: This statement has been added accordingly.

Response: Ethics statement has been appeared only in the Method section (Lines 112-118).

Response to Reviewer1 (R1)

Reviewer #1: The manuscript presents a comprehensive evaluation of the Elecsys® HCV Duo immunoassay, highlighting its potential as a diagnostic tool for Hepatitis C virus (HCV) infection. The study offers valuable insights into the assay's performance, particularly in its ability to detect both HCV antibodies and core antigen. However, there are several critical aspects that need to be addressed to enhance the manuscript's clarity, the robustness of the findings, and the practical implications of the assay. Below are specific comments and concerns that should be addressed before the manuscript can be considered for publication.

Minor Comments:

1. Ensure consistent tense usage. For example, "This study aimed to evaluate" could be revised to "This study aims to evaluate" to match the present tense used throughout the abstract.

Response: This has been revised accordingly. (line 28)

2. There are a few instances where commas are either missing or incorrectly placed. For example, in the sentence "The results were interpreted as reactive for Duo, as calculated from the highest score among anti-HCV and HCV cAg results," a comma after "Duo" seems unnecessary.

Response: This Sentence has been rewritten in the revied version.

3. The manuscript might benefit from ensuring consistency in the use of italics and bold for emphasis. For instance, "HCV" is sometimes italicized and other times not. Consistency is key.

Response: All terms used in this version have been corrected for consistency.

4. Ensure that all acronyms are defined at their first occurrence. For example, "ECLIA" and "COI" are used without prior definition. 

Response: We have rechecked all acronyms that have been mentioned at the first met.

5. The resolution of the figures is very poor. Consider using color coding to differentiate between the different groups in figure 1. This would help visually separate different parts of the flow and make the figure more engaging.

Response: All figures have been reconstructed for consistency and readability, and we have implemented the color coding as per your recommendation. All figures have been converted by the PACE generator software to correct format before resubmission. 

Major comments

1. One of the primary concerns with this manuscript is that it does not substantially add to the body of knowledge already established by the 2022 study on the Elecsys® HCV Duo immunoassay. The earlier study provided a comprehensive multicenter evaluation of the assay's specificity and sensitivity across diverse populations and settings. While your manuscript offers localized data from a Thai cohort, the overall findings align closely with those of the 2022 paper, without introducing significant new insights or advancements. To strengthen the manuscript, the authors should highlight the novelty of their study and what their study add to the exiting literature.

Response: Some additional paragraphs were included to address the study's gap (Line 74-79), and we are continuing the investigation of DUO/cAg testing by using the Duo-assay in clinical samples. The 2022 study focused on blood donors and routine clinical samples, calculating sensitivity and specificity based on 31 out of 32 positive samples (Table 3, DOI: 10.1016/j.jcv.2022.105293). In contrast, our study expands upon this by calculating these metrics from a larger cohort of actively infected and resolved cases.

Moreover, we extend to investigate the performance using samples from individuals with active phase and those who recovered from HCV infection to monitor the recurrent HCV infection. We also performed the correlation between Duo/HCVcAg and HCV RNA and also deeply investigated whether different genotypes of HCV affect the performance of Duo/HCVcAg detection. (lines 105-107)

2. The manuscript briefly mentions the lack of correlation between HCV core antigen (cAg) levels and HCV RNA. This is an important point, but it needs a more in-depth analysis. The discussion should explore potential reasons for this lack of correlation and its implications for the use of the Elecsys® HCV Duo immunoassay as a stand alone diagnostic tool. 

Response 2.1 Notably, we found no correlation between the DUO/HCVcAg assay and RT-PCR, as discussed in detail (Lines Line 308-318). Previous studies predominantly examined this correlation using the Architect HCVcAg assay (Abbott)

Our previous study also found the correlation between cAg (Architect) and RT-PCR 

“Hansoongnern P, Pratedrat P, Nilyanimit P, Wasitthankasem R, Posuwan N, Wanlapakorn N, et al. An amino acid substitution in HCV core antigen limits its use as a reliable measure of HCV infection compared with HCV RNA. PLOS ONE. 2023;18:e0287694. “

Additionally, compare these findings with the 2022 study, which did not emphasize this issue, and discuss whether this observation is novel or aligns with other emerging data in the field.

Response 2.2: We ensured that our manuscript presents novel findings. The 2022 study primarily involved blood donors and routine samples, most of which were from healthy (non-infected) individuals. In contrast, our current study evaluates the performance of the DUO-assay (Elecsys, Roche) in samples from infected, resolved infection, and non-infected individuals. 

Additionally, we are the first to report the influence of genotypes on this assay (Elecsys HCV Duo immunoassay, Roche). (Lines 308-310) 

3. The manuscript explicitly states that there was no overall correlation found between HCV core antigen (cAg) levels and HCV RNA. Given this finding, the rationale for proceeding with genotyping and sequencing is unclear. The authors need to explain why they chose to perform genotyping after determining that there was no significant correlation between cAg and RNA. Was the genotyping intended to investigate whether specific HCV genotypes might influence this lack of correlation, or was there another hypothesis being tested? Without this explanation, the inclusion of genotyping and sequencing appears disconnected from the primary findings, raising questions about the necessity and relevance of these procedures within the study. This clarification is crucial for understanding the study's design and the implications of the results.

Response: 

Firstly, we tried to study the correlation from overall samples between cAg and HCV RNA levels (among 1a, 1b, 3a and 6f). The result showed the no significant and negligible positive correlation (R-squared = 0.06730, p<0.0001). Then, to identify deeply if we analyze upon different genotype, this will appear that are there any relationship among cAg and HCV or not. However, the result showed it still has no correlation between cAg and HCV RNA for respective genotypes. 

we expected that the genotype might influence to cAg detection due to the previous study in Hansoongnern et. al, 2022 showed the relation between specific genotype when used the architect cAg assay. However, our studies showed that the Elecsys DUO/cAg has no correlated with RT-PCR result. The reason had been described in discussion (lines 308-318)

“Hansoongnern P, Pratedrat P, Nilyanimit P, Wasitthankasem R, Posuwan N, Wanlapakorn N, et al. An amino acid substitution in HCV core antigen limits its use as a reliable measure of HCV infection compared with HCV RNA. PLOS ONE. 2023;18:e0287694. “

4. The manuscript should reference any existing literature that suggests genotyping might influence the correlation between cAg and RNA levels. If such references exist, they should be cited to justify the decision to include genotyping in the study. If no such references are available, the authors should clarify that their approach was exploratory and explain the reasoning behind this exploration. This clarification is crucial for understanding the study's design and the implications of the results.

Response: We appreciated on your comment. However, do same change in the revised version. Although HCV RNA detection was a gold standard for diagnosis of HCV infection, the performance of HCV RNA comparing with Architect HCVcAg (Abbott GmbH) might be affected by the genotype diversity (a). In addition, the study of correlation between HCV RNA and Architect HCVcAg antigen (identified using CMIA (Abbott)) in different HCV genotype demonstrated a positive correlation between HCV RNA and HCV cAg was observed in only genotype 1b (b). Therefore, we decided to analyze the correlation between HCV cAg and HCV RNA in our study. The review literature for genotyping might influence the correlation between cAg and RNA levels have been cited in lines 95-99. (introduction) 

(a) “Hansoongnern P, Pratedrat P, Nilyanimit P, Wasitthankasem R, Posuwan N, Wanlapakorn N, et al. An amino acid substitution in HCV core antigen limits its use as a reliable measure of HCV infection compared with HCV RNA. PLOS ONE. 2023;18:e0287694. “

(b) “Xiang Y, Lai XF, Chen P, Yang Y. The correlation of HCV RNA and HCV core antigen in different genotypes of HCV. J Clin Lab Anal. 2018;33:e22632.” https://pmc.ncbi.nlm.nih.gov/articles/PMC6430366/

5. The manuscript reports that a significant number of samples in the active infection group tested positive for anti-HCV but negative for HCV cAg. This finding is critical, as it impacts the reliability of using cAg as a standalone marker for active infection. The authors should discuss potential reasons for this discrepancy, such as assay sensitivity, low viral loads, or technical factors. They should also address the clinical implications of these findings, particularly the importance of confirming anti-HCV+/cAg- results with HCV RNA testing to avoid false negatives.

Response: I agree with your comment regarding the lower sensitivity of the Duo/HCVcAg assay. However, WHO guidelines do permit the use of cAg as a marker for evaluating infection and monitoring treatment response in active cases. Evidence from previous studies using the Architect assay has demonstrated its successful application in clinical samples with high viral loads, while showing lower sensitivity in samples with low viral loads. In many high-income countries with advanced healthcare systems, RT-PCR is still regarded as the gold standard for all samples. This suggests that RT-PCR remains necessary to confirm results in anti-HCV+/cAg- cases (This has been mentioned in discussion Lines 370-373 and conclusion line 395-403).

Response to Reviewer2 (R1)

Reviewer #2: Reviewer Comments to Authors: Manuscript No.: PONE-D-24-35203

Comparison of Elecsys® HCV Duo assay versus Anti-HCV RDT followed by qRT-PCR to Identify active infection

I went through the manuscript, and after thoroughly reviewing and evaluating the scientific attributes and

level of importance of the manuscript, I am hereby sending the following comments to the Authors:

The authors’ aim was to compare the Elecsys® HCV Duo immunoassay and the traditional two-step

process, which involves initial anti-HCV antibody testing followed by quantitative real-time polymerase

chain reaction (qRT-PCR) for detecting active hepatitis C infections. Additionally, the study aimed to

evaluate the relationship between HCVcAg and qRT-PCR assay in different genotypes.

Comments to Authors:

1. While the study holds significant value, especially within the context of the global effort to

eliminate HCV, the authors should consider the following points:

A. Introduction:

i. In line 69: include a reference on the adoption of HCV core antigen testing by WHO

recommendation. This will provide credibility regarding the global significance of core antigen

testing.

Response: This has been added in lines 72-74.

 ii. Provide a brief mention of the national guidelines for HCV testing in Thailand and whether

HCV core antigen testing has been adopted in these guidelines. This will help ground the study

in the context of local diagnostic practices and highlight the relevance of your findings for

Thailand’s national guidelines.

Response: This has been provided in lines 74-79.

iii. Elaborate on the advantages of using the assay of anti-HCV combined with HCV cAg in real

world settings, particularly its implications for reducing costs and turnaround time in HCV

screening and diagnostics.

Response: We have mentioned the advantages of the use of Duo (combination between anti-HCV/HCVcAg) in Disccussion (Lines 356-359) and conclusion section (Lines 399-401)

B. Materials and Methods:

i. Flow and Clarity: Improving the flow of the methodology would significantly enhance its

clarity and make it easier for readers to follow. A more logical structure would ensure that

each step of the study is clearly outlined, facilitating comprehension. For example, as noted

below, you could improve the transition between the previous and current study methods for

better readability.

Response: We have rewritten the serology assay section to clarify the flow of methods and revised Figure 1 diagram to clarify the transition between the previous and current study and highlighted the samples from “active infection group”, “Resolved infection group”, and “Non-infected group” by different colors. 

ii. Serology assay in the Laboratory assessment Section: Reorganize this section by starting with

information on how the samples were categorized into active infection and resolved infection

groups, referencing the use of Bioline™ HCV RDT from the previous study (Figure 1). Then, follow this by explaining how the current study employed the Elecsys® Duo assay (anti-HCV & HCV cAg) to further assess these groups. This approach will make it easier for readers to understand how the two testing methods are related and integrated into the study.

Response: This section has been reorganized for clarity in study design and sample collection section.

iii. Sample Processing: More detail should be mentioned on how samples were processed, in

particular in regard to the handling of frozen samples and any potential effects on antigen or

RNA stability. 

Response: We mentioned the sampl

---

## [Editor Report · Decision Letter 1]

31 Oct 2024

Comparison of anti-HCV combined with HCVcAg (Elecsys® HCV Duo immunoassay) and anti-HCV rapid test followed by HCV RNA analysis using qRT-PCR to identify active infection for treatment

PONE-D-24-35203R1

Dear Prof. Poovorawn, 

We’re pleased to inform you that your manuscript has been judged scientifically suitable for publication and will be formally accepted for publication once it meets all outstanding technical requirements. One small comment: change the very last sentence last sentence in the abstract FROM "However, an additional qRT-PCR step is required to confirm the diagnosis in the case of anti-HCV+/HCVcAg– samples TO : Nevertheless, cases with anti-HCV+/HCVcAg– results should undergo additional confirmation with western blot/immunoblot and qRT-PCR to ensure diagnostic accuracy, especially in Blood donation facilities"

Kind regards,

Gheyath K. Nasrallah

Academic Editor

PLOS ONE

Additional Editor Comments (optional):

One small comment: change the very last sentence last sentence in the abstract FROM "However, an additional qRT-PCR step is required to confirm the diagnosis in the case of anti-HCV+/HCVcAg– samples TO : Nevertheless, cases with anti-HCV+/HCVcAg– results should undergo additional confirmation with western blot/immunoblot and qRT-PCR to ensure diagnostic accuracy, especially in Blood donation facilities"
---

## [Editor Report · Acceptance letter]

7 Nov 2024

PONE-D-24-35203R1 

PLOS ONE

Dear Dr. Poovorawan, 

I'm pleased to inform you that your manuscript has been deemed suitable for publication in PLOS ONE. Congratulations! Your manuscript is now being handed over to our production team.

Kind regards, 

on behalf of

Dr. Gheyath K. Nasrallah 

Academic Editor

PLOS ONE